# Genome-Wide Association Screens for Anterior Cruciate Ligament Tears

**DOI:** 10.3390/jcm13082330

**Published:** 2024-04-17

**Authors:** Vincenzo Candela, Umile Giuseppe Longo, Alessandra Berton, Giuseppe Salvatore, Francisco Forriol, Alessandro de Sire, Vincenzo Denaro

**Affiliations:** 1Fondazione Policlinico Universitario Campus Bio-Medico, Via Alvaro del Portillo 200, 00128 Rome, Italy; v.candela@policlinicocampus.it (V.C.); a.berton@policlinicocampus.it (A.B.); g.salvatore@policlinicocampus.it (G.S.); denaro.cbm@gmail.com (V.D.); 2Research Unit of Orthopaedic and Trauma Surgery, Department of Medicine and Surgery, Università Campus Bio-Medico di Roma, Via Alvaro del Portillo 21, 00128 Rome, Italy; 3Orthopaedic Surgery Department, Hospital Universitario Puerta de Hierro, 28222 Majadahonda, Madrid, Spain; submission.cbm@gmail.com; 4Physical and Rehabilitative Medicine Division, Department of Experimental and Clinical Medicine, University of Catanzaro “Magna Graecia”, 88100 Catanzaro, Italy; alessandro.desire@unicz.it; 5Research Center on Musculoskeletal Health, MusculoSkeletalHealth@UMG, University of Catanzaro “Magna Graecia”, 88100 Catanzaro, Italy

**Keywords:** anterior cruciate ligament, ACL, rupture, genetics, single-nucleotide polymorphisms, SNPs

## Abstract

Background: The etiopathogenesis of ACL rupture is not clarified. The aim of this study is to identify genomic regions and genetic variants relevant to anterior cruciate ligament injury susceptibility that could be involved in non-contact anterior cruciate ligament ruptures. Methods: A systematic review of the literature was performed according to the Preferred Reporting Items for Systematic Reviews and Meta-Analyses (PRISMA) guidelines with a PRISMA checklist and algorithm. A search of PubMed, MEDLINE, CINAHL, Cochrane, EMBASE, and Google Scholar databases was conducted using combinations of the terms “anterior cruciate ligament”, “ACL”, “rupture”, “genetics”, “single nucleotide polymorphisms”, and “SNP” since the inception of the databases until 2021. Results: Twenty-three studies were included. A total of 7724 patients were analyzed. In total, 3477 patients had ACL ruptures and 4247 patients were controls. Genetic variants in genes encoding for collagens, elastin, fibrillin, matrix metalloproteinases, proteoglycans, angiogenesis-associated signaling cascade proteins, growth differentiation factors, tissue inhibitors of metalloproteases, interleukins, and fibrinogen were analyzed. Conclusion: Findings regarding the association between genes encoding for collagen (COL3A1, COL1A1, and COL12A1), aggrecan (ACAN), decorin (DCN), matrix metalloproteinase (MMP3), interleukin 6 (IL-6), vascular endothelial growth factor A (VEGFA), biglycan (BGN), fibrinogen (FGB), and ACL injuries were found to be inconclusive. Additional evidence is required in order to establish substantial conclusions regarding the association between genetic variants and ACL rupture.

## 1. Introduction

The anterior cruciate ligament (ACL) is a stable static structure made up of a large number of collagen fibers. The ACL is composed of the anteromedial (AM) bundle and the posterolateral (PL) bundle. When the knee is flexed, the AM bundle is tightened and the PL bundle is untightened; on the other hand, when the knee is extended, the PL bundle is tightened. The PL bundle also limits internal and external rotation [1].

Collagen is the main component of the ACL ligament. The remaining components are proteoglycans, elastin, and other proteins and glycoproteins [1].

ACL rupture is a serious injury in sports and produces pain, joint effusion, and reduced functional performance. On the other side, in the long term, ACL rupture leads to knee osteoarthritis [2].

The etiopathogenesis of ACL rupture is not clarified [3]. ACL rupture could be caused by non-contact events, direct contact events, or indirect contact events.

Non-contact ACL rupture refers to a type of injury to the anterior cruciate ligament (ACL) of the knee that occurs without direct external force applied to the knee joint. Instead, it typically transpires during sudden pivoting, cutting, or landing movements, often associated with sports activities such as basketball, soccer, or skiing. This type of injury occurs when the knee undergoes rapid deceleration or changes direction abruptly, placing excessive stress on the ACL, leading to its tearing or rupture. Non-contact ACL ruptures are significant due to their prevalence among athletes and their potential for prolonged recovery periods and associated complications, including instability and the increased risk of future knee injuries [4,5,6,7]. Several studies have indicated correlations between ACL rupture and different genetic variations, thereby potentially indicating that genetic predisposition plays a significant role in non-contact ACL rupture [4,5,6,7].

Genetic factors can exert influence on anterior cruciate ligament (ACL) injuries through various mechanisms. Firstly, certain genetic variations may predispose individuals to altered the structural integrity or strength of ligaments, including the ACL, rendering them more susceptible to injury even under minimal stress. Additionally, genetic factors can impact neuromuscular control, proprioception, and biomechanics, affecting how individuals move and land during physical activities. Variations in collagen composition, which influences ligament strength and elasticity, may also be genetically determined, contributing to ACL injury risk. Furthermore, genetic predispositions may interact with environmental factors, such as training intensity or biomechanical stresses, exacerbating the susceptibility to ACL injuries [4,5,6,7].

Regarding genetic factors, single-nucleotide polymorphisms (SNPs) of genes encoding for collagens, proteoglycans, matrix metalloproteinases, angiogenesis-associated signaling cascades and growth differentiation hormone factors, elastin and fibrillin, and interleukins play a role in causing non-contact ACL ruptures.

1.Genes encoding for collagen

Collagen, being the primary structural protein in ligaments, provides tensile strength and structural integrity to the ACL. Variations in genes responsible for encoding collagen can lead to alterations in the structure, composition, and mechanical properties of the ligament, thereby impacting its susceptibility to injury [8].

Specifically, genetic variations in genes encoding for collagen types I, III, V, and XII have been implicated in ACL injuries [8]. Collagen type I is predominant in ligaments and contributes to their mechanical strength. Mutations or alterations in the *COL1A1* gene, which encodes for collagen type I, can affect the overall strength and integrity of the ACL, potentially predisposing individuals to ligament tears [8].

Collagen type III, encoded by the *COL3A1* gene, is involved in fibrillogenesis and the formation of collagen fibers. Genetic variations in COL3A1 may disrupt the normal assembly and organization of collagen fibrils within the ACL, leading to structural weaknesses and increased vulnerability to injury.

Collagen type V, encoded by the *COL5A1* gene, plays a role in regulating the diameter of collagen fibrils and forming heterotypic fibrils with collagen type I. Variations in COL5A1 may disrupt the proper interaction between collagen types I and V, affecting the mechanical properties of the ACL and potentially increasing the risk of injury.

Additionally, collagen type XII, encoded by the *COL12A1* gene, is involved in regulating the organization and mechanical properties of collagen fibril bundles. Genetic alterations in COL12A1 may impair the structural organization of collagen within the ACL, compromising its strength and stability [9].

2.Genes encoding for proteoglycans

Proteoglycans are essential components of the extracellular matrix (ECM) within ligaments, including the ACL, contributing to its structural integrity, hydration, and biomechanical properties. Several genes encoding for proteoglycans, such as decorin (DCN), biglycan (BGN), and fibromodulin (FMOD), play crucial roles in modulating the ECM composition and organization within the ACL [10]. These proteoglycans interact with collagen fibrils and other matrix components, regulating fibrillogenesis, the diameter of fibrils, and tissue architecture. Decorin, encoded by the *DCN* gene, is a small proteoglycan known for its ability to control the diameter of collagen fibrils during fibrillogenesis. Mutations or alterations in the *DCN* gene may disrupt the normal organization of collagen fibrils within the ACL, compromising its structural integrity and predisposing individuals to ligament injuries.

Similarly, biglycan (encoded by the *BGN* gene) and fibromodulin (encoded by the *FMOD* gene) are proteoglycans that modulate collagen assembly and organization within the ECM. Variations in these genes can affect the interactions between proteoglycans and collagen fibrils, predisposition plays a significant role in non-contact ACL rupture. Furthermore, proteoglycans are involved in maintaining tissue hydration and lubrication, which are crucial for the proper function and resilience of ligaments like the ACL. Disruptions in proteoglycan synthesis or function due to genetic variations may compromise the hydration and lubrication of the ACL, rendering it more susceptible to mechanical stress and injury [11].

3.Genes encoding for matrix metalloproteinases

MMPs are a family of enzymes responsible for degrading components of the ECM.

In the context of ACL injuries, MMPs are implicated in the remodeling and degradation of ECM components within the ligament. Specifically, MMPs can degrade collagen fibers, weakening the structural integrity of the ACL and rendering it more susceptible to injury. Additionally, MMPs may disrupt the balance between ECM synthesis and degradation, leading to abnormal tissue remodeling and impaired healing responses following injury. Genetic variations in genes encoding for MMPs, such as MMP1, MMP3, MMP10, and MMP12, can modulate the activity and expression levels of these enzymes [12]. Dysregulated MMP activity due to genetic variations can disrupt the structural integrity and biomechanical properties of the ACL, increasing the risk of injury and impairing the healing response following damage. Overall, genes encoding for matrix metalloproteinases play a critical role in modulating ECM remodeling processes within the ACL and can influence its susceptibility to injury [12,13,14].

4.Genes encoding for angiogenesis-associated signaling cascades and growth differentiation hormone factors

The angiogenesis-associated signaling cascade encompasses various genes and pathways involved in the formation of new blood vessels, a process known as angiogenesis. Angiogenesis plays a crucial role in tissue repair and regeneration, including in the ACL. Genetic variations in genes encoding for angiogenic factors, such as vascular endothelial growth factor A (VEGFA), can impact the angiogenic response within the ACL following injury. Dysregulated angiogenesis due to genetic variations may impair the vascularization and healing processes of the ACL, leading to delayed or inadequate repair and increased susceptibility to reinjury [15,16,17].

Similarly, growth differentiation hormone factors, such as the growth differentiation factor 5 (GDF5), are involved in regulating cell growth, differentiation, and tissue repair. Genetic variations in genes encoding for GDF factors can affect their expression levels and signaling pathways, influencing the remodeling and repair processes within the ACL. Dysregulated GDF signaling due to genetic variations may disrupt the normal tissue homeostasis and repair mechanisms of the ACL, rendering it more prone to injury and impaired healing responses [15,16,17].

5.Genes encoding for elastin and fibrillin

Elastin is a protein that contributes to the elasticity and flexibility of connective tissues, including ligaments like the ACL. Genetic variations in genes encoding for elastin, such as the *ELN* gene, can impact the structural integrity and biomechanical properties of the ACL. Alterations in elastin synthesis or structure due to genetic variations may compromise the elasticity and resilience of the ACL, increasing its vulnerability to injury [18].

Fibrillin is a glycoprotein that forms microfibrils in the extracellular matrix, providing structural support and stability to tissues. Genetic variations in genes encoding for fibrillin, such as the *FBN1* and *FBN2* genes, can affect the assembly and organization of fibrillin microfibrils within the ACL. Disruptions in fibrillin synthesis or function due to genetic variations may weaken the structural integrity of the ACL, making it more prone to mechanical stress and injury [18].

6.Genes encoding for interleukins

Interleukins are a group of cytokines that play critical roles in modulating immune responses, inflammation, and tissue repair processes. Several interleukins, such as IL-1, IL-6, and IL-10, have been implicated in the pathophysiology of ACL injuries.

IL-1 is known for its pro-inflammatory properties and can promote the production of matrix metalloproteinases (MMPs) and other inflammatory mediators that contribute to tissue degradation and remodeling [19]. Genetic variations in genes encoding for IL-1, such as the *IL1A*, *IL1B*, and *IL1RN* genes, can influence the expression levels and activity of IL-1, thereby affecting the inflammatory response within the ACL following injury. IL-6 is another cytokine involved in inflammation and tissue repair processes. It can stimulate the production of collagen and other extracellular matrix components, promoting tissue healing and regeneration. Genetic variations in genes encoding for IL-6, such as the *IL6* gene, may impact the levels of IL-6 production and signaling, influencing the repair processes within the ACL.

IL-10, on the other hand, is an anti-inflammatory cytokine that can suppress the production of pro-inflammatory mediators and modulate immune responses. Genetic variations in genes encoding for IL-10, such as the *IL10* gene, can affect the anti-inflammatory properties of IL-10, potentially influencing the resolution of inflammation and tissue healing processes within the ACL [19].

7.Genes encoding for b-fibrinogen

Beta-fibrinogen is a component of the blood clotting cascade and plays a crucial role in the formation of fibrin clots, which are essential for wound healing and tissue repair processes [20]. Genetic variations in genes encoding for b-fibrinogen, such as the FGB gene, can affect the structure, function, and levels of fibrinogen circulating in the blood. Elevated levels of fibrinogen have been associated with increased blood viscosity, impaired microcirculation, and enhanced inflammatory responses. These effects can contribute to the development of a pro-thrombotic state and endothelial dysfunction, which may increase the risk of vascular complications and tissue damage, including within the ACL [20].

Genome-wide association studies (GWASs) could be helpful in carefully relating genetic information to patients’ clinical situation. GWASs offer an examination of different allele frequencies of a genome-wide set of selected polymorphic genetic variants (usually single-nucleotide polymorphisms; SNPs) in different groups of individuals to see if any variant is associated with a specific phenotypic trait. Such studies are particularly useful in finding genetic variants that contribute to individual susceptibility to common complex diseases. This approach enables high-powered whole-genome association and grants consistently high coverage across different populations, maximizing genome power and discovery rates.

Therefore, the aim of this systematic review was to identify genomic regions and genetic variants relevant to ACL injury susceptibility that could be involved in non-contact ACL ruptures.

## 2. Materials and Methods

A systematic review of the literature was performed according to the Preferred Reporting Items for Systematic Reviews and Meta-Analyses (PRISMA) guidelines with a PRISMA checklist and algorithm (Figure 1). A search of PubMed, MEDLINE, CINAHL, Cochrane, EMBASE, and Google Scholar databases was conducted using combinations of the terms “anterior cruciate ligament”, “ACL”, “rupture”, “genetics”, “single nucleotide polymorphisms”, and “SNP”. The search was conducted separately by three independent reviewers (U.G.L., V.C., and F.F.). Initially, all articles were checked for relevance. To avoid selection bias and errors, the three investigators independently assessed each publication’s abstract. The last search was performed on 25 September 2021. Articles in English and case–control studies were included. Articles that reported the association between genetic variants and ACL rupture in both the ACL and control groups were included. Literature reviews; case reports; studies on animals; studied on cadavers or in vitro; biomechanical reports; technical notes; letters to editors; and instructional courses were excluded. All investigators independently extracted the following data: the design of the study; the number of patients; the genes involved; and the association with ACL injuries.

Three independent reviewers (U.G.L., V.C., and A.B.) conducted a risk-of-bias assessment of the studies using the Cochrane Centre’s ‘case–control tool’ [21]. Any discrepancies were resolved through consensus analysis. This tool comprised six questions, four of which addressed bias (refer to Table 1). Selection bias was evaluated based on the recruitment source for cases and controls, with an ideal comparison involving population-based controls. Confounding was assessed considering age and sex, with optimal scenarios involving matched or adjusted groups. Furthermore, studies were evaluated for information bias, aiming for consistency in DNA extraction and genotype determination methods. The risk of bias was categorized into three levels: low, high, and unclear. A study was deemed ‘high-risk’ if at least one bias question received a ‘no’ response and ‘low-risk’ if all other questions received a ‘yes’ response. Studies were labeled ‘unclear risk’ if all questions received a ‘doubtful’ response or a combination of ‘doubtful’ and ‘yes’.

## 3. Results

### 3.1. Study Selection

A total of 260 studies were found when the search results from all databases were combined. Twenty studies were duplicates so they were removed. After screening the title and abstract, a total of 212 investigations were eliminated. After reading the complete text, four studies were eliminated (two because they did not look at the link between genetic variants and ACL ruptures and two because they were not written in full). In the end, 24 publications were included in this systematic review. Figure 1 depicts a flowchart of this procedure.

### 3.2. Study Characteristics

A summary of the 24 included studies is shown in Table 2. All the included studies were case–control studies. Twelve studies examined genetic variants in or near genes encoding collagens; four examined matrix metalloproteinases; three examined proteoglycans; one study examined a variant near growth–differentiation; four genes examined the angiogenesis-associated signaling cascade; and, finally, elastin and fibrillin, interleukins, and b-fibrinogen were analyzed in one study each.

A total of 7724 patients were analyzed; 3477 patients had ACL ruptures and 4247 patients were controls.

a.
**Collagen**


The gene most frequently examined was COL1A1. Conflicting findings emerged regarding the association between the TT and GG genotypes of the COL1A1 rs1800012 variant and ACL rupture. Likewise, conflicting evidence was noted regarding the association between the AA genotype of the COL3A1 rs1800255 variant and ACL rupture. Zhao et al. [20] suggest that males with the rs970547 A allele and the rs970547 AA genotype of COL12A1 may be at high risk for developing an ACL rupture. Poor evidence was obtained, suggesting that no association between the COL5A1 rs12722 and COL5A1 rs13946 variants was identified. Furthermore, the evidence was deemed insufficient to establish an association between COL1A1 rs1107946 and COL6A1 rs35796750 variants and ACL rupture.

b.
**Proteoglycans**


Conflicting findings have emerged regarding the association between the ACAN rs1516797, BGN rs1042103, and BGN rs1126499 variants and ACL rupture. Mannion et al. [11] found that ACAN rs1516797 was independently associated with the risk of developing ACL injury in all analyzed participants. Cięszczyk et al. [10] showed that the ACAN rs1516797 G/T genotype was under-represented in the control group compared to the male ACL injury group. Moreover, they found that the BGN rs1042103 A allele was significantly under-represented in the male control group compared to the male ACL injury group. Willard et al. [22] found that allele combinations across BGN (rs1126499 and rs1042103), COL5A1 (rs12722), and DCN (rs516115), as well as eight miRNA recognition sequences, are involved in increasing susceptibility to ACL rupture.

Evidence indicating an association between the GG (protective) genotype of the DCN rs516115 variant and ACL rupture was inadequate. Similarly, evidence establishing an association between the DCN rs13312816, ACAN rs2351491, ACAN rs1042631, FMOD rs7543148, FMOD rs10800912, and LUM rs2268578 variants and ACL rupture was insufficient.

c.
**Matrix Metalloproteinases**


Lulinska-Kuklik et al. found that the MMP3 rs591058C and rs679620 G alleles [23] were associated with ACL rupture. Inadequate evidence was found to support the absence of any association between the MMP1 rs1799750, MMP3 rs679620, MMP3-1612, MMP10 rs486055, and MMP12 rs2276109 variants and ACL rupture.

d.
**Angiogenesis-Associated Signaling Cascades and Growth Differentiation Hormone Factors**


Conflicting findings have emerged regarding the association between VEFGA rs699947 and ACL rupture. Shukla et al. [24] demonstrated that the VEGF A allele (rs699947) and I allele (rs35569394) were significantly over-represented in the ACL injury group. Rahim et al. [16] and Cięszczyk et al. [10] did not find the same association for rs699947.

Evidence indicating an association between the GA genotype (deleterious) of the VEGFA rs1570360 variant and ACL rupture was inadequate. Additionally, evidence indicating the association between the VEFGA rs2010963, KDR 1870377, KDR rs2071559, NGFB rs6678788, HIF1A rs11549465, and GDF5 rs143383 variants and ACL rupture was insufficient.

e.
**Elastin and Fibrillin**


Evidence indicating no association between the ELN rs2071307 variant and ACL rupture or no association between FBN2 rs331079 variant and ACL rupture was insufficient.

f.
**Interleukins**


Lulinska-Kuklik et al. found that rs1800795 *IL6* gene polymorphism [19] was associated with ACL rupture.

g.
**B-fibrinogen**


rs1800787, rs1800788, rs1800790, and rs2227389 genotypes in the b-fib promoter region were found to be associated with ACL injury.

**Table 2 jcm-13-02330-t002:** Characteristics of the included studies.

Study	Year	Design	Patients with ACL Ruptures (n)	Controls (n)	Gene	Product	Variant	Association with ACL Ruptures (Differences in Patients with ACL Ruptures vs. Control)
Ficek et al. [25]	2013	Case– control	91	143	COL1A1	Collagen	rs1800012 and rs1107946	No significant differences
Ficek et al. [26]	2014	Case– control	91	143	COL12A1	Collagen	rs970547	No significant differences
Khoschnau et a [27]	2008	Case– control	233	325	COL1A1	Collagen	rs1800012	No significant differences
Khoury et al. [18]	2015	Case– control	141	219	ELN	Elastin	rs2071307	No significant differences
-	-	-	-	-	FBN2	Fibrillin-2	rs331079	No significant differences
Malila et al. [28]	2011	Case– control	86	100	MMP3	Matrix metalloproteinase	–1612	No significant differences
Mannion et al. [11]	2014	Case– control	227	234	ACAN	Proteoglycans	rs2351491, rs1042631, and rs1516797	rs2351491 and rs1042631: no significant differences **ACAN rs1516797: was significantly under-represented in the controls group (*p* = 0.024)**
	-	-	-	-	BGN		rs1126499 and rs1042103	No significant differences
	-	-	-	-	DCN		rs13312816 and rs516115	rs13312816: no significant differences; **rs516115: significant association**
	-	-	-	-	FMOD		rs7543148 and rs10800912	No significant differences
	-	-	-	-	LUM		rs2268578	No significant differences
O’Connell et al. [29]	2015	Case– control	242 (South African population) 91 (Polish population)	235 (South African population) 91 (Polish population)	COL3A1 and COL6A1	Collagen	rs1800255 and rs35796750	**rs1800255; significant association**; rs35796750: no significant differences
Posthumus et al. [30]	2009	Case– control	117	130	COL1A1	Collagen	rs1800012	**rs1800012: significant association**
Posthumus et al. [31]	2009	Case– control	129	216	COL5A1	Collagen	rs12722 and rs13946	No significant differences
Posthumus et al. [32]	2010	Case– control	129	216	COL12A1	Collagen	rs970547 and rs240736	No significant differences
Posthumus et al. [14]	2012	Case– control	129	216	MMP1, MMP3, MMP10, and MMP12	Matrix metalloproteinase	rs1799750, rs679620, rs486055, and rs2276109	No significant differences
Rahim et al. [16]	2014	Case– control	227	227	VEGFA, KDR, NGFB, and HIF1A	Angiogenesis-associated signaling cascade genes	rs699947, rs1570360, rs2010963, rs1870377, rs2071559, rs6678788, and rs11549465	No significant differences
Raleigh et al. [17]	2013	Case– control	126	216	GDF5	Growth differentiation factor	rs143383	No significant differences
Stepien-Słodkowska et al. [33]	2013	Case– control	138	183	COL1A1	Collagen	rs1800012	**Significant association**
Stepien-Słodkowska et al. [34]	2015	Case– control	138	183	COL3A1	Collagen	rs1800255	**Significant association**
Stepien-Słodkowska et al. [35]	2015	Case– control	138	183	COL3A1	Collagen	rs13946 and rs12722	No significant differences
Cięszczyk et al. [10]	2017	Case– control	143	229	ACAN	Proteoglycans	rs1516797	**Significant association** T/T (*p* > 0.041)
					BGN		rs1042103 and rs1126499	**BGN rs1042103 A allele: significant association in the male**
					DCN		rs516115	No significant differences
					VEGFA	Angiogenesis-associated signaling cascade genes	rs699947	No significant differences
Lulinska-Kuklik et al. [23]	2018	Case– control	229	192	MMP3	Matrix metalloproteinases	rs591058C/T and rs679620 G/A	**Significant association:** MMP3 rs591058C and rs679620 G alleles were significantly over-represented in cases compared to controls (*p* = 0.021)
					MMP8	Matrix metalloproteinases	rs11225395C/T	No significant differences
					TIMP2	Tissue inhibitors of metalloproteases	rs4789932 G/A	No significant differences
Lulinska et al. [13]	2020	Case– control	228	202	MMP1	Matrix metalloproteinases	rs1799750, →G	No significant differences
					MMP10	Matrix metalloproteinases	rs486055 C > T	No significant differences
					MMP12	Matrix metalloproteinases	rs2276109 T > C	No significant differences
Lulinska-Kuklik et al. [19]	2019	Case– control	229	194	IL1B	Interleukins	rs16944 and rs1143627	No significant differences
					IL6	Interleukins	rs1800795	**Significant association**
					IL6R	Interleukins	rs2228145	No significant differences
Shukla et al. [24]	2020	Case– control	90	76	VEGFA	Angiogenesis-associated signaling cascade genes	rs699947 and rs35569394	**Significant association:** the A allele (rs699947) and I allele (rs35569394) were significantly over-represented in the ACL group
Shukla et al. [36]	2020	Case– control	90	76	COLIA1	Angiogenesis-associated signaling cascade genes	COLIA1 Sp1 + 1245 G > T	No significant differences
Willard et al. [22]	2017	Case– control	227	234	BGN	Proteoglycans	rs1126499 and rs1042103	**Significant association**
					COL5A1	Collagen	rs12722	**Significant association**
					DCN	Proteoglycans	rs516115	**Significant association**
Zhao et al. [20]	2020	Case– control	101	110	COL1A1	Collagen	rs1800012	No significant differences
					COL5A1	Collagen	rs12722 and rs13946	No significant differences
					COL12A1	Collagen	rs970547 and rs240736	**Significant association**
					FGB	B-fibrinogen	rs1800787, rs1800788, rs1800789, rs1800790, rs1800791, and rs2227389	rs1800789 and rs1800791: no significant differences; rs1800787, rs1800788, rs1800790, and rs2227389: **significant association;** **rs1800787, rs1800788, rs1800790, and rs2227389 genotypes in** **the b-fib promoter region: association with ACL injuries**

### 3.3. Risk of Bias

Risk of bias is shown in Table 3. In the risk-of-bias assessment, nine studies were determined to have a high risk of bias. Additionally, 15 studies were categorized as unclear in terms of bias risk.

## 4. Discussion

ACL ruptures are very common in athletes and their instability could lead to an increased risk of future knee injuries, thus making a rehabilitative prevention plan necessary [37,38,39,40].

In this systematic review, we investigated the genetic factors that could predispose athletes to ACL ruptures through simple familial predisposition studies and complex gene association studies. Brothers and sisters (siblings) of patients with a history of ACL ruptures seem to have susceptibility to developing an anterior cruciate ligament rupture. Various authors, by analyzing patients with ACL ruptures who underwent ACL reconstructive surgery from 2010 to 2013, found that having a sibling with a history of ACL ruptures and being female are important risk factors for developing ACL ruptures in the contralateral knee [40]. Flynn et al. [5] reported that having a familial history of ACL ruptures in first-, second-, and third-degree relatives can double the risk of developing ACL ruptures compared to patients without a relative with a history of ACL ruptures. Those data reflect the genetic background of ACL rupture. However, nowadays, the relationship between genetic factors and ACL ruptures or other musculoskeletal injuries is not clearly understood.

The existing literature on genetic variations associated with ACL rupture is summarized in this comprehensive review that includes 24 studies. In the 24 articles that we analyzed, conflicting evidence on the effect of different gene variants on ACL rupture was found.

The most important genes that we analyzed are as follows: COL3A1 [29,34], COL1A1 [30,33], COL12A1 [20], ACAN [10,11], DCN [11,22], MMP3 [23], IL6 [19], VEGFA [24], BGN [22], and FGB [20].

The analyzed studies exhibited either an unclear or high risk of bias and confounding factors. In the risk-of-bias assessment, nine studies were determined to have a high risk of bias. Additionally, 15 studies were categorized as unclear in terms of bias risk. This underscores the importance of evaluating the quality of research and the implications for drawing conclusive findings.

Moreover, we observed heterogeneity in the genetic variants under investigation, outcome definitions, and genetic contrasts, precluding the feasibility of conducting a formal meta-analysis. Consequently, we conducted a best-evidence synthesis of the current literature.

Variants of the analyzed genes were found to play a role in the synthesis, strength, and homeostasis of the ligament. COL1A1 encodes collagen type I, which imparts mechanical strength to various tissues, including ligaments. COL3A1 encodes collagen type III and participates in the fibrillogenesis of collagen type I. The type I collagen molecule is a heterotrimer made up of two a1 chains and one a2 chains, and it is the most abundant protein in ligaments, accounting for 70–80% of their dry weight. The collagen Iα1 (COLIA1) and the collagen Iα2 (COLIA2) genes are the two primary genes that control collagen synthesis.

A polymorphism in the promoter region of COLIA1 intron 1, which is a putative binding site for the transcription factor Sp1, has been linked to several illnesses [41]. The substitution of thymine (T) for guanine (G) in COLIA1 intron 1 is known as “s”, whereas “SS” indicates homozygosity for GG.

Conflicting findings have emerged regarding the association between the TT and GG genotypes of the COL1A1 rs1800012 variant and ACL rupture. Likewise, conflicting evidence has been noted regarding the association between the AA genotype of the COL3A1 rs1800255 variant and ACL rupture. COL5A1 encodes collagen type V, which interacts with collagen type I in the formation of heterotypic fibrils and modulates the diameter of these fibrils. Collagen type XII, encoded by the *COL12A1* gene, constitutes the largest member among fibril-associated collagens, regulating the organization and mechanical properties of collagen fibril bundles. Decorin, encoded by the *DCN* gene, is a member of the small proteoglycan family and plays a role in restricting the diameter of collagen fibrils during fibrillogenesis.

MMPs have a role in the breakdown of the extracellular matrix in both normal physiological processes (like embryonic development, reproduction, and tissue remodeling), as well as pathological events (like arthritis and metastasis).

Interleukins (ILs) play essential roles in the activation and differentiation of immune cells, as well as in proliferation, maturation, migration, and adhesion. They also have pro-inflammatory and anti-inflammatory properties.

The protein encoded by the fibrinogen beta chain (FGB) gene is the beta component of fibrinogen, a blood-borne glycoprotein comprising three pairs of nonidentical polypeptide chains. It is involved in hemostasis and antimicrobial host defense.

VEGFA, which encodes the vascular endothelial growth factor A, serves as a regulator of angiogenesis.

The precise implications of the above-mentioned genetic variants remain uncertain.

Our included studies presented several limitations.

We did not impose a minimum sample size threshold to ensure the inclusion of all available studies, given that genetic investigations typically necessitate larger participant cohorts than those featured in most of our included studies. However, the sample sizes became notably small when stratifying groups based on sex or the mechanism of injury, thereby precluding the reporting of any stratified analyses.

Moreover, several genes and variants were examined in the same population group. While both of these genetic variants may have an association with the risk of developing an ACL rupture, it is conceivable that only one of them truly contributes to the risk, while the association of the other genetic variant is potentially confounding.

The idea of using genetic tests for screening athletes predisposed to sports-related injuries is an appealing one. Genetic testing could help to avoid injury, especially in particular sports. Genetic information could also be used by clinicians and athletes to make more informed decisions regarding LCA injury prevention, diagnosis, and management.

Nevertheless, additional data are essential in order to establish a conclusive association between genetic variants and ACL rupture. Augmented sample sizes and more comprehensive genetic investigations are imperative in order to attain a thorough understanding of these potential associations.

## 5. Conclusions

Taken together, the findings of this systematic review showed that the association between the COL3A1, COL1A1, COL12A1, ACAN, DCN, MMP3, IL6, VEGFA, BGN, and FGB genes and ACL injuries should be considered inconclusive. More evidence is needed to draw significant conclusions regarding the association between genetic variants and ACL rupture.

## Figures and Tables

**Figure 1 jcm-13-02330-f001:**
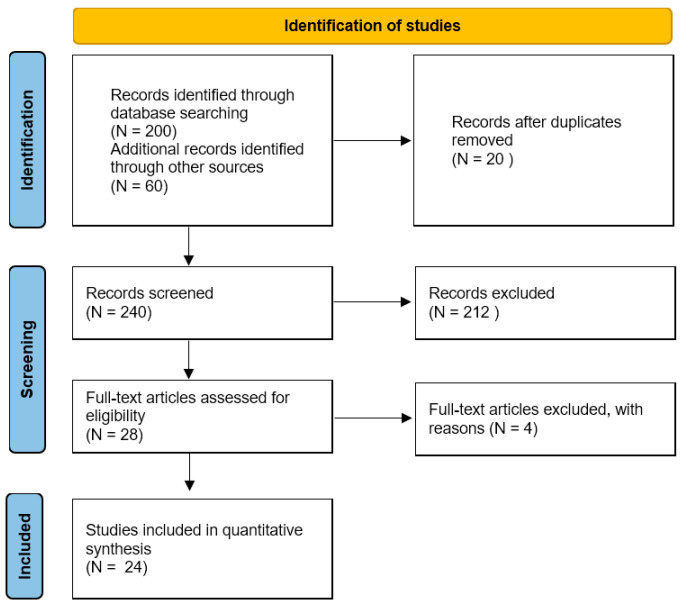
PRISMA flowchart.

**Table 1 jcm-13-02330-t001:** Risk of bias questions.

Criterion	Question
Case	Have the cases been clearly and sufficiently defined?
Control	Have the controls been clearly and adequately defined?
Selection bias	Has selection bias been adequately addressed and excluded?
Defined exposure	Has the exposure been clearly defined, and is the method employed to assess this exposure deemed appropriate?
Determination	Was blinding to exposure status maintained prior to determining the presence of the disease?
Confounding	Have the primary confounding factors been identified and adequately accounted for in both the study design and analysis?

**Table 3 jcm-13-02330-t003:** Risk of bias.

	Case	Control	Selection Bias	Defined Eposure	Determination Exposure	Confouding	Overall
Ficek et al. [25]	+	+	?	+	?	+	?
Ficek et al. [26]	+	+	?	+	?	+	?
Khoschnau et al. [27]	+	?	+	+	+	-	-
Khoury et al. [18]	+	+	?	+	?	-	-
Malila et al. [28]	+	+	?	+	?	-	-
Mannion et al. [11]	+	+	?	+	?	-	-
O’Connell et al. [29]	+	?	?	+	+	-	-
Posthumus et al. [30]	+	+	?	+	?	-	-
Posthumus et al. [31]	+	+	?	+	?	+	?
Posthumus et al. [32]	+	+	?	+	?	+	?
Posthumus et al. [14]	+	+	?	+	?	+	?
Rahim et al. [16]	+	?	?	+	?	-	-
Raleigh et al. [17]	+	+	?	+	?	+	?
Stepien-Słodkowska et al. [33]	+	+	?	+	?	?	?
Stepien-Słodkowska et al. [34]	+	+	?	+	?	?	?
Stepien-Słodkowska et al. [35]	+	+	?	+	?	?	?
Cięszczyk et al. [10]	+	+	?	+	?	?	?
Lulinska-Kuklik et al. [23]	+	+	?	+	?	+	?
Lulinska et al. [13]	+	+	?	+	?	+	?
Lulinska-Kuklik et al. [19]	+	+	?	+	?	+	?
Shukla et al. [24]	+	+	?	+	?	-	-
Shukla et al. [36]	+	+	?	+	?	?	?
Willard et al. [22]	+	+	?	+	?	+	?
Zhao et al. [20]	+	+	?	+	?	-	-

## Data Availability

The datasets used and/or analyzed during the current study are available from the corresponding author on reasonable request.

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
