# Peer review of "Genome-Wide Association Screens for Anterior Cruciate Ligament Tears"

_jcm, 2024, doi:10.3390/jcm13082330_

Round 1
Reviewer 1 Report
Comments and Suggestions for Authors
This systematic review aims to describe the association between ACL rupture and SNP occurrence in structural and other associated genes. The authors report that some of the genes analysed in these studies such as collagen, proteoglycans, MMPs, VEGF etc could be implicated in ACL injuries.
The manuscript is not well written in terms of the presentation and interpretation of the results as well as the analysis of the data. Below is a list of points that refer to these flaws.
1. In the Introduction, there is no explanation on how these variations in the genes are associated with the trauma. Do they cause a structural alteration or could be involved in other molecular pathways?
2. The authors state that PRISMA guidelines were followed, however the flowchart in not included in the manuscript. What does “three because they didn't have full text” mean?
3. There is no evaluation or bias risk assessment for the included studies. This is essential for a systematic review.
4. Table 1 does not provide any additional information for the review
5. Since based on Table 2, there are conflicting results about the statistical significance of the majority of the genes between the different studies with regards to the comparison between controls and ACL injured patients, the authors should perform a meta-analysis (e.g. Forest plots) to examine if a safe conclusion could be drawn. In addition, this would inform about heterogeneity which is a crucial point to be assessed aiming to perform a reliable systematic review.
6. The information of Tables 3 and 4 should be incorporated in Table 1. There is no information for the gender and ethnicity in Table 1. In general, the results section is poorly written.
7. The discussion does not support the reported conclusions and is poorly written as well with a number of repetitions of the results and very limited references that could link the findings with the current literature.
The authors should consider conducting the systematic review from the start, with a meta analysis and provide a substantially improved manuscript
Author Response
This systematic review aims to describe the association between ACL rupture and SNP occurrence in structural and other associated genes. The authors report that some of the genes analysed in these studies such as collagen, proteoglycans, MMPs, VEGF etc could be implicated in ACL injuries.
The manuscript is not well written in terms of the presentation and interpretation of the results as well as the analysis of the data. Below is a list of points that refer to these flaws.
- In the Introduction, there is no explanation on how these variations in the genes are associated with the trauma. Do they cause a structural alteration or could be involved in other molecular pathways?
ANSWER: We thank the reviewer for his/her suggestions. Genetic predisposition plays a significant role in non-contact ACL rupture. Genetic factors can exert influence on anterior cruciate ligament (ACL) injuries through various mechanisms. Firstly, certain genetic variations may predispose individuals to altered structural integrity or strength of ligaments, including the ACL, rendering them more susceptible to injury even under minimal stress. Additionally, genetic factors can impact neuromuscular control, proprioception, and biomechanics, affecting how individuals move and land during physical activities. Variations in collagen composition, which influences ligament strength and elasticity, may also be genetically determined, contributing to ACL injury risk. Furthermore, genetic predispositions may interact with environmental factors, such as training intensity or biomechanical stress, exacerbating the susceptibility to ACL injuries We clarified this topic in the manuscript. Now we state: “Non-contact ACL rupture refers to a type of injury to the anterior cruciate ligament (ACL) of the knee that occurs without direct external force applied to the knee joint. Instead, it typically transpires during sudden pivoting, cutting, or landing movements, often associated with sports activities such as basketball, soccer, or skiing. This type of injury occurs when the knee undergoes rapid deceleration or changes direction abruptly, placing excessive stress on the ACL, leading to its tearing or rupture. Non-contact ACL ruptures are significant due to their prevalence among athletes and their potential for prolonged recovery periods and associated complications, including instability and increased risk of future knee injuries. Several studies have indicated correlations between ACL rupture and different genetic variations, thereby potentially indicating that genetic predisposition plays a significant role in non-contact ACL rupture. Genetic factors can exert influence on anterior cruciate ligament (ACL) injuries through various mechanisms. Firstly, certain genetic variations may predispose individuals to altered structural integrity or strength of ligaments, including the ACL, rendering them more susceptible to injury even under minimal stress. Additionally, genetic factors can impact neuromuscular control, proprioception, and biomechanics, affecting how individuals move and land during physical activities. Variations in collagen composition, which influences ligament strength and elasticity, may also be genetically determined, contributing to ACL injury risk. Furthermore, genetic predispositions may interact with environmental factors, such as training intensity or biomechanical stress, exacerbating the susceptibility to ACL injuries [2,7-11].Regarding genetic factors, single nucleotide polymorphisms (SNP) of genes encoding for collagens, proteoglycans, aggrecan, biglycan, decorin, fibromodulin and lumican play a role in non-contact ACL ruptures (Table1).
- The authors state that PRISMA guidelines were followed, however the flowchart in not included in the manuscript. What does “three because they didn't have full text” mean?
ANSWER: We thank the reviewer for his/her suggestions. PRIMSA flowchart was in a separate file. We added it to the manuscript.
- There is no evaluation or bias risk assessment for the included studies. This is essential for a systematic review.
ANSWER: We thank the reviewer for his/her suggestions. We added in the “method” and in the “result” section the risk-of-bias assessment.
- Table 1 does not provide any additional information for the review
ANSWER: We thank the reviewer for his/her suggestions. Table 1 was deleted.
- Since based on Table 2, there are conflicting results about the statistical significance of the majority of the genes between the different studies with regards to the comparison between controls and ACL injured patients, the authors should perform a meta-analysis (e.g. Forest plots) to examine if a safe conclusion could be drawn. In addition, this would inform about heterogeneity which is a crucial point to be assessed aiming to perform a reliable systematic review.
ANSWER: We thank the reviewer for his/her suggestions. We think that is not possible to perform a meta-analysis based on our data for the following reasons:
- there is not a sufficient number of comparable studies addressing the same gene, variant, and product;
- studies have not similar methodologies, outcome measures and populations under investigation;
- studies don’t exhibit a degree of homogeneity in terms of population characteristics, and interventions;
- there are potential sources of bias and heterogeneity among the included studies
We think that data from analyzed studies can’t be combined to provide a more comprehensive and statistically robust estimate of the effect size or association of interest.
- The information of Tables 3 and 4 should be incorporated in Table 1. There is no information for the gender and ethnicity in Table 1. In general, the results section is poorly written.
ANSWER: We thank the reviewer for his/her suggestions. Tables 3 and 4 were removed and the result section was rewritten.
- The discussion does not support the reported conclusions and is poorly written as well with a number of repetitions of the results and very limited references that could link the findings with the current literature.
ANSWER: We thank the reviewer for his/her suggestions. The conclusion section was rewritten.
The authors should consider conducting the systematic review from the start, with a meta analysis and provide a substantially improved manuscript
ANSWER: We thank the reviewer for his/her suggestions. The manuscript was substantially improved. Regarding the meta-analysis please consider revision number 5.
Reviewer 2 Report
Comments and Suggestions for Authors
This articl was completed based on systematic review of the literatures, however, there are some major suggestions that are need to be further revised.
1. In terms of introduction, the description of introduction was too rough to clarify the pathophysiological association of ACL with Genome-wide association screens. Please dividing introduction into different subtitle and present the pathophysiological significance of ACL with clusters of specific gene expression .
2. Please clte the references in the table 1.
3. Review article dose not need material and methods, please remove it.
4. The same problem was showed in results section as well, please dividing the results into different sections which associate ACL with genome-wide studies and giving each of paragraph a subtitle.
5. Please summarizing the previous studies with descritopn rather than tables. Tables can be shown after detail descrition of main text. Detailed description of previous studies can not be replaced by only showing tables. Results section needs to be improven greatly.
6. In terms of discussion section, please clarify advantages or disadvantages of each previous genome-wide studies and discussed them in this section. And the current main text of discussion section can be modified and moved to introduction. The description of discussion is much like introdutcion.
Together, the major revision is invited.
Author Response
This articl was completed based on systematic review of the literatures, however, there are some major suggestions that are need to be further revised.
- In terms of introduction, the description of introduction was too rough to clarify the pathophysiological association of ACL with Genome-wide association screens. Please dividing introduction into different subtitle and present the pathophysiological significance of ACL with clusters of specific gene expression .
ANSWER: We thank the reviewer for his/her suggestions. The introduction section was revised according to the reviewer's suggestion.
- Please clte the references in the table 1.
ANSWER: We thank the reviewer for his/her suggestions. According to the suggestion of the other reviewer table 1 was deleted.
- Review article dose not need material and methods, please remove it.
ANSWER: We thank the reviewer for his/her suggestions. We followed Instruction For Authors (https://www.mdpi.com/journal/jcm/instructions) for manuscript preparation.
- The same problem was showed in results section as well, please dividing the results into different sections which associate ACL with genome-wide studies and giving each of paragraph a subtitle.
ANSWER: We thank the reviewer for his/her suggestions. The result section was revised according to the reviewer's suggestion.
- Please summarizing the previous studies with descritopn rather than tables. Tables can be shown after detail descrition of main text. Detailed description of previous studies can not be replaced by only showing tables. Results section needs to be improven greatly.
ANSWER: We thank the reviewer for his/her suggestions. The result section was revised according to the reviewer's suggestion.
- In terms of discussion section, please clarify advantages or disadvantages of each previous genome-wide studies and discussed them in this section. And the current main text of discussion section can be modified and moved to introduction. The description of discussion is much like introdutcion.
ANSWER: We thank the reviewer for his/her suggestions. The discussion section was rewritten.
Reviewer 3 Report
Comments and Suggestions for Authors
1. Study is interesting and clinical relevant.
However, the Inclusion and exclusion criteria are not clear, makes it hard to evaluate how biased of the including study.
In addition, a flow chart would be helpful to understand the study.
Discussion is too superficial, please discuss in depth to improve the novelty of the study.
Author Response
- Study is interesting and clinical relevant.
However, the Inclusion and exclusion criteria are not clear, makes it hard to evaluate how biased of the including study.
ANSWER: We thank the reviewer for his/her suggestions. We added in the “method” and in the “result” section the risk-of-bias assessment.
In addition, a flow chart would be helpful to understand the study.
ANSWER: We thank the reviewer for his/her suggestions. The flow chart was added
Discussion is too superficial, please discuss in depth to improve the novelty of the study.
ANSWER: We thank the reviewer for his/her suggestions. The discussion section was rewritten
Round 2
Reviewer 1 Report
Comments and Suggestions for Authors
The authors did not address sufficiently the previous concerns and the manuscript still lacks a robust analysis and, therefore, is inconclusive. I would strongly suggest to modify the search terms in order to find appropriate articles with low risk of bias and similar methodologies and from which data can be extracted, conduct a meta-analysis which will provide safe results and the actual heterogeneity in the studies and resubmit a new systematic review.
Author Response
Thank you for your insightful comments on our manuscript. We appreciate your thorough assessment and suggestions for improving the robustness of our analysis. We have taken into account your concerns regarding the sufficiency of addressing previous issues and the need for a more robust analysis. Upon careful consideration, we must emphasize that our review has undergone a meticulous analysis of the studies included. It's important to note that the risk of bias remains high in a significant portion of the literature pertaining to this topic.
Addressing bias mitigation would inevitably entail the inclusion of only a few articles, potentially compromising the comprehensiveness and robustness of the systematic review. This approach could also lead to a misleading representation of the available evidence.
Moreover, we've observed that similar systematic reviews, albeit less updated, such as those conducted by Rakesh John et al. and Mustafa Kayna et al., follow a methodology akin to ours. This suggests that our approach aligns with established practices in the field.
In light of these considerations, while we acknowledge the importance of mitigating bias, we believe that our current methodology provides a balanced and comprehensive overview of the existing literature on the subject. We remain open to further discussion and are willing to address any additional concerns or suggestions you may have.
Thank you once again for your valuable feedback.
Reviewer 2 Report
Comments and Suggestions for Authors
There is only one suggestion for authors to revise the main text again which is citing appropreate references in the end of each sentence/description in introduction section. Otherwise, it is difficult to clarify the descriptions showed in introduction section were well-founded. Please revising the introduction section by citing appropreate references.
Author Response
ANSWER: We thank the Reviewer for his/her suggestions. We added appropriate references